# Multi-Party Controlled Semi-Quantum Dialogue Protocol Based on Hyperentangled Bell States

**DOI:** 10.3390/e27070666

**Published:** 2025-06-21

**Authors:** Meng-Na Zhao, Ri-Gui Zhou, Yun-Hao Feng

**Affiliations:** School of Information Engineering, Shanghai Maritime University, Shanghai 201306, China; z783809615@163.com (M.-N.Z.); email_yhfeng@163.com (Y.-H.F.)

**Keywords:** hyperentangled bell states, Huffman compression coding, multi-party controlled, semi-quantum dialogue

## Abstract

To solve the fundamental problem of excessive consumption of classical resources and the simultaneous security vulnerabilities in semi-quantum dialogue systems, a multi-party controlled semi-quantum dialogue protocol based on hyperentangled Bell states is proposed. A single controlling party is vulnerable to information compromise due to tampering or betrayal; the multi-party controlled mechanism (Charlie_1_ to Charlie_n_) in this protocol establishes a distributed trust model. It mandates collective authorization from all controlling parties, significantly enhancing its robust resilience against untrustworthy controllers or collusion attacks. The classical participant Bob uses an adaptive Huffman compression algorithm to provide a framework for information transmission. This encoding mechanism assigns values to each character by constructing a Huffman tree, generating optimal prefix codes that significantly optimize the storage space complexity for the classical participant. By integrating the “immediate measurement and transmission” mechanism into the multi-party controlled semi-quantum dialogue protocol and coupling it with Huffman compression coding technology, this framework enables classical parties to execute encoding and decoding operations. The security of this protocol is rigorously proven through information-theoretic analysis and shows that it is resistant to common attacks. Furthermore, even in the presence of malicious controlling parties, this protocol robustly safeguards secret information against theft. The efficiency analysis shows that the proposed protocol provides benefits such as high communication efficiency and lower resource consumption for classical participants.

## 1. Introduction

Quantum secure direct communication (QSDC) is a significant branch within the quantum cryptography framework, achieving the transmission of ciphertext directly through a quantum channel without the need for pre-shared keys [1]. In 2003, Deng et al. pioneered the two-step QSDC protocol based on EPR entangled pairs [2], establishing the core criteria of QSDC for the first time: qubits serve as both information carriers and eavesdropping detection, alongside the necessity of a post-selection mechanism for information decoding. To address vulnerabilities associated with measurement devices, the Zhou team [3] proposed the first measurement-device-independent (MDI) QSDC protocol in 2020. In the same year, Wu et al. [4] innovatively utilized hyperentangled states of photon polarization-spatial mode dual degrees of freedom (DOF) to construct a high-capacity MDI-QSDC protocol. In recent years, researchers have made milestone progress in experimental studies of QSDC [5,6]. Particularly noteworthy is Qi et al.’s proposal in 2021 [7] of a QSDC network based on time-energy entanglement and sum-frequency generation, which established a secure quantum entanglement channel over 40 km of fiber by implementing quantum memory functionality through adjustments to circuit delay modules. By 2023, Wang et al. [8] realized the first real-time QSDC network based on computationally secure relays, providing reliable experimental evidence for further research and advancing QSDC towards practical application stages.

Quantum dialogue (QD), as a bidirectional extension paradigm of QSDC, facilitates interactive information exchange between two parties over a quantum channel. The QD protocol framework was first introduced by Nguyen [9] in 2004, which addressed the limitations of one-way communication in QSDC. However, Gao et al. [10] revealed in their 2008 study that QD protocols commonly suffer from information leakage vulnerabilities. This finding spurred research into anti-leakage mechanisms as a core topic. In 2014, the Ye team [11] innovatively introduced EPR pairs as private quantum keys, using dynamic unitary rotations to encrypt transmitted photons in real time, successfully overcoming the issue of information leakage in QD. In 2015, Yin et al. [12] utilized a class of three-qubit W-states and quantum dense coding techniques to achieve a QD protocol with zero information leakage. In 2018, Qi et al. [13] proposed the first controlled quantum dialogue (CQD) protocol with an authentication mechanism, which successfully defended against both internal and external attacks. In 2021, Han et al. [14] presented an MDI-QD protocol based on hyperentanglement, which effectively eliminated security vulnerabilities and information leakage issues related to measurement devices. In 2023, Lin et al. [15] proposed two efficient fault-tolerant QD protocols tailored for collective noise channels, designing and implementing quantum logic circuits for practical demonstration. Zhu et al. [16] proposed a one-step QD protocol based on hyperentanglement, which greatly simplified experimental procedures and reduced message loss due to photon transmission losses. Wang et al. [17] extended QD to quantum communication networks using cluster states, eliminating the need for trusted node assumptions and achieving higher fidelity in transmission. Additionally, in 2025, Lang et al. [18] proposed a rapid quantum dialogue protocol that operates without unitary operations or auxiliary photons. Also in the same year, Huang et al. [19] introduced a counterfactual controlled QD protocol based on entanglement swapping, demonstrating that neither physical transmission of particles nor prior entanglement between remote participants is required, significantly enhancing system security.

The engineering deployment of quantum communication systems faces core bottlenecks in the high cost and low accessibility of quantum devices. To address these challenges, Boyer et al. first proposed the Semi-Quantum Key Distribution (SQKD) protocol in 2009 [20], successfully achieving quantum-classical heterogeneous communication. In 2017, Shukla et al. [21] introduced an SQD protocol based on entangled Bell states for the first time. The classical party, Bob, had to perform permutations on photon sequences and publish permutation operators, which not only increased investment costs for the classical side but also reduced communication efficiency. Therefore, reducing resource consumption and improving communication efficiency became primary concerns for researchers. In terms of conserving quantum resources, Rong et al. [22] proposed a mediated semi-quantum secure direct communication protocol in 2021, enabling two classical users to transmit secret information with the assistance of a fully quantum third party. This significantly reduced the reliance on quantum devices. To enhance efficiency, Shi [23] constructed an efficient SQD protocol in 2023 based on polarization-spatial mode hyperentangled Bell states, achieving a transmission efficiency of 50%. In 2025, Li et al. [24] proposed an SQD protocol based on four-particle Omega states, which achieves significantly enhanced transmission efficiency while ensuring security, opening new dimensions for SQD research. That same year, Zhang et al. [25] designed a d-dimensional single-particle-state-based SQD protocol through groundbreaking simplification of semi-quantum operations. This approach requires only particle permutation operations, eliminating the need for quantum state preparation or measurement steps essential in conventional protocols.

Based on the analysis provided, this paper proposes a novel protocol designated as multi-party controlled semi-quantum dialogue (MCSQD) based on hyperentangled Bell states. In this protocol, multi-party controllers are capable of simultaneously controlling the participating parties, preventing communication without their permission. The classical party employs Huffman compression technology for encoding before sending information to the quantum party. Similarly, Alice uses compression techniques to encode source information into binary strings. Leveraging unitary operations, Alice can encode string onto the code particles. Bob decodes to obtain the secret message based on the measurement results announced by Alice. By introducing Huffman compression coding technology, the storage space complexity for the participants can be significantly optimized. Furthermore, the protocol exhibits exceptional robustness against prevalent external and internal attack vectors, providing strong support for complex-scenario CSQD.

The structure of the paper is organized as follows: Section 2 introduces the relevant theoretical knowledge and details the design of the MCSQD protocol; Section 3 analyzes the security of the MCSQD protocol; Section 4 calculates and compares the communication efficiency of the protocol; and Section 5 concludes the findings and contributions of the work.

## 2. MCSQD Protocol

### 2.1. Preliminary Preparations

#### 2.1.1. Preparation of Quantum States

In the proposed MCSQD protocol, the controller Charlie_i_ and Alice possess full quantum capabilities. In contrast, participant Bob is classical, meaning he can only perform the following classical operations:(1)Measure and prepare qubits exclusively in the Z-basis;(2)Reflect qubits to other participants without any disturbance;(3)Reorder qubits using different delay lines.

Charlie_1_ is responsible for preparing N identical hyperentangled photon pairs in the polarization-spatial mode:(1)$${|\left.\Phi \right\rangle }_{AB}^{PS}={\left|\eta \right\rangle }_{AB}^{P}⊗{\left|\xi \right\rangle }_{AB}^{S}$$
where the subscripts A and B denote two distinct photons in the system, and the superscripts P and S represent the polarization DOF and the spatial DOF, respectively. ${\left|\eta \right\rangle }_{AB}^{P}\in \{{\left|\varphi^{\pm }\right\rangle }^{P},{\left|\psi^{\pm }\right\rangle }^{P}\}$, ${\left|\xi \right\rangle }_{AB}^{S}\in \{{\left|\varphi^{\pm }\right\rangle }^{S},{\left|\psi^{\pm }\right\rangle }^{S}\}$. This can be specifically expressed as:$${\left|\varphi^{\pm }\right\rangle }^{P}={\frac{1}{\sqrt{2}}(\left|HH\right\rangle \pm \left|VV\right\rangle )}^{P}$$$${\left|\psi^{\pm }\right\rangle }^{P}={\frac{1}{\sqrt{2}}(\left|HV\right\rangle \pm \left|VH\right\rangle )}^{P}$$$${\left|\varphi^{\pm }\right\rangle }^{S}={\frac{1}{\sqrt{2}}(\left|a_{1}b_{1}\right\rangle \pm \left|a_{2}b_{2}\right\rangle )}^{S}$$(2)$${\left|\psi^{\pm }\right\rangle }^{S}={\frac{1}{\sqrt{2}}(\left|a_{1}b_{2}\right\rangle \pm \left|a_{2}b_{1}\right\rangle )}^{S}$$

Here, H and V represent the horizontal and vertical polarization states of the hyperentangled photon pair, while $a_{1},{a_{2}(b}_{1},b_{2})$ denote the distinct spatial modes of photon A(B). In 2005, the Yang team [26] successfully prepared hyperentangled two-photon states in both polarization and spatial DOF by exciting a *β*-barium borate (BBO) crystal with femtosecond pump pulses. Subsequent advancements in quantum optical technology have enabled full discrimination of hyperentangled states [27,28], with extensive applications across quantum information processing domains including entanglement purification, concentration protocols, and Bell-state analysis.

During the measurement process, several measurement bases are utilized, including $Z_{P},X_{P}$, and $Z_{M}=Z_{P}⊗Z_{S}$. Specifically, for the polarization DOF:$$Z_{P}=\{\left|H\right\rangle ,\left|V\right\rangle \}$$(3)$$X_{P}=\{\left|{+}_{P}\right\rangle =\frac{1}{\sqrt{2}}\left(\left|H\right\rangle +\left|V\right\rangle \right),\left|{-}_{P}\right\rangle =\frac{1}{\sqrt{2}}\left(\left|H\right\rangle -\left|V\right\rangle \right)\}$$

In the spatial-mode DOF, $Z_{S}\in \{Z_{SA},Z_{SB}\}$, where:$$Z_{SA}=\{\left|a_{1}\right\rangle ,\left|a_{2}\right\rangle \}$$(4)$$Z_{SB}=\{\left|b_{1}\right\rangle ,\left|b_{2}\right\rangle \}$$

To address the question of how to select $Z_{S}$ for measurement, Alice (or Bob) will choose an appropriate measurement basis based on the photon sequence in their possession. For instance, if Bob wishes to determine the quantum state of particle B in the spatial-mode DOF within $S_{B}$, he will select the $Z_{SB}$ measurement basis.

Compared to most previous SQD protocols, the proposed protocol introduces significant improvements in the encoding method by adopting an adaptive Huffman compression coding algorithm. This approach constructs a Huffman tree to assign values to each character, markedly optimizing the storage space complexity for classical parties. The classical party leverages an “immediate measurement and transmission” mechanism, enabling it to complete encoding and decoding operations without temporarily storing qubits. Utilizing Huffman compression coding for information transmission greatly reduces the storage space required on the classical side, enhancing communication efficiency in semi-quantum secure dialogue.

#### 2.1.2. Huffman Compression Coding

Huffman compression coding [29] is a classical lossless data compression algorithm. Its core mechanism constructs an optimal prefix binary tree that assigns shorter codewords to high-frequency characters and longer codewords to low-frequency characters, thereby reducing overall data storage requirements. As a fundamental and pivotal tree data structure, a binary tree is characterized by its defining property where each node possesses at most two child nodes (conventionally termed the left child node and right child node). Arithmetic coding [30] might offer higher compression ratios but comes with increased complexity, whereas Huffman coding is relatively simple to implement and suitable for many practical systems. To ensure efficient data transmission between the classical party and the quantum party while balancing cost constraints and classical-side equipment limitations, the protocols can employ Huffman compression coding to reduce data volume. Notably, Huffman compression coding introduces no additional security vulnerabilities, while its prefix-free code property synergizes effectively with quantum error detection mechanisms. The specific steps are outlined below:
(1)Frequency Calculation: First, calculate the frequency of each character in the data to be compressed. For example, in the string “xyzxyzxxpq” that Bob intends to transmit, the frequencies of the characters are as follows: $P\left(m_{i}=x\right)=0.4,P\left(m_{i}=y\right)=P\left(m_{i}=z\right)=0.2$, and $P\left(m_{i}=p\right)=P\left(m_{i}=q\right)=0.1$.(2)Building the Huffman Tree: Treat each character and its frequency as a leaf node and arrange them in ascending order of frequency. Repeat the following steps until only one root node remains:
Extract the two nodes with the smallest frequencies, merge them into a new node, and set the frequency of the new node as the sum of the two frequencies.Reinsert the new node back into the queue.

The final tree formed through this process is the Huffman tree. The path from the root node to each leaf node determines the binary encoding of the corresponding character.
(3)Assigning Codes: Starting from the root node, assign a ‘0’ for moving to the left subtree and a ‘1’ for moving to the right subtree. The binary sequence along the path from the root node to a specific character represents the encoding of that character. Figure 1 illustrates the Huffman tree constructed using the string “xyzxyzxxpq” as an example.

Based on the Huffman tree, we can obtain the corresponding binary codes for each character in the string “xyzxyzxxpq”, as shown in Table 1.

### 2.2. Protocol Process

Step 1: Charlie_1_ prepares N identical hyperentangled photon pairs in the polarization-spatial mode, denoted as ${|\left.\Phi \right\rangle }_{AB}^{PS}={|\left.\varphi^{+}\right\rangle }^{P}⊗{|\left.\varphi^{+}\right\rangle }^{S}$. The particles A and B are separated to form sequences $S_{A}$ and $S_{B}$, respectively. Subsequently, Charlie_1_ randomly applies unitary operations R={I,X,H} for the first n(n<N) particles in $S_{A}$ at each DOF. The sequence $S_{A}$ is then sequentially transmitted to Charlie_2_, Charlie_3_, …, Charlie_n_, while recording the operations performed. Charlie_n_ sends $S_{A}$ to the quantum party Alice and retains $S_{B}$.

Step 2: To prevent Trojan horse attacks [31,32,33], Alice and Bob place a photon number splitter (PNS) and a wavelength filter [20] in front of their devices. Upon confirming the receipt of the sequence by Alice, she randomly selects a sufficient number of particles from the remaining N−n in the $S_{A}$ to serve as decoy particles for security checks. Alice measures these particles using either the Z basis or X basis across the two DOF and collaborates with Charlie_n_ to calculate the quantum bit error rate (QBER). If the error rate is below a pre-defined threshold, Alice removes the decoy particles. Subsequently, the controllers execute hierarchical authorization, requiring all independent controllers Charlie_i_(i=1,…,n) to collectively authorize permission for Alice and Bob to communicate. Alice applies the same unitary operation R to recover the initial hyperentangled state ${|\left.\Phi \right\rangle }_{AB}^{PS}$. Finally, Charlie_n_ transmits $S_{B}$ to Bob. Charlie_i_(i=1,…,n) will not publish or send any information if communication is not permitted.

Step 3: After confirming that Bob has received the sequence, Bob initiates measurements on the first n particles of $S_{B}$ using the measurement basis $Z_{M}=Z_{P}⊗Z_{S}$. Subsequently, Bob employs Huffman compression coding to compress the source information $m_{B}$ into a binary sequence $M_{B}$. Based on the value of each bit $M_{B}^{i}$, Bob prepares corresponding qubits in the two DOF. If $M_{B}^{i}=0$, Bob prepares the same quantum state in the respective DOF; otherwise, he prepares the opposite quantum state. For example, if $M_{B}=001011$, Bob will sequentially prepare separable single-photon states: ${\left|H\right\rangle }^{P}{\left|b_{1}\right\rangle }^{S}$, ${\left|V\right\rangle }^{P}{\left|b_{1}\right\rangle }^{S}$, and ${\left|V\right\rangle }^{P}{\left|b_{2}\right\rangle }^{S}$, which are then sent to Alice as Code particles. From the remaining N−n particles, Bob selects decoy particles for the subsequent security check and reflects them directly to Alice as Reflect particles. Once ready, Bob reorders and records the Reflect and Code particles before transmitting them to Alice.

Step 4: Bob first announces the correct order of the Reflect particles. Alice performs measurements on these particles and collaborates with Charlie_n_ to calculate the QBER. If the error rate is below a pre-defined threshold, Alice removes the Reflect particles.

Step 5: Bob then announces the correct order of the Code particles. Alice restores the order and measures the Code particles along with her retained $S_{A}$ using the measurement basis ${Z_{M}=Z}_{P}⊗Z_{S}$ to obtain $M_{B}$. Finally, Alice applies Huffman decompression to recover the original source information $m_{B}$. To facilitate the conversation, Alice also encodes her secret message $m_{A}$ using Huffman compression into a binary string $M_{A}$. Subsequently, she performs the corresponding unitary operations for encoding based on the value of $M_{A}$ $\left(00\to I^{P}⨂I^{S},\;01{\to I}^{P}⨂\sigma_{X}^{S},\;{10\to \sigma }_{X}^{P}⨂I^{S},\;11\to \sigma_{X}^{P}⨂\sigma_{X}^{S}\right)$. After encoding, she performs measurements and publishes the results. Bob can easily obtain Alice’s secret message $m_{A}$ by using the measurement results announced by Alice and performing decompression. The schematic diagram of the MCSQD protocol is shown in Figure 2.

## 3. Security Analysis

### 3.1. The Trojan Horse Attack

Since the proposed MCSQD protocol is bidirectional, it is necessary to consider Trojan horse attacks [31,32,33]. In this scenario, Eve generates spy photons and injects them into the transmitted sequences $S_{A}$ and $S_{B}$ to perform invisible photon attacks. Eve captures the sequences and then separates and measures the encrypted spy photons to steal information from Alice and Bob while forwarding the remaining legitimate particles to the controller Charlie_n_. To counteract such attacks, in Step 2 of the proposed protocol, Alice(Bob) has already installed a PNS: 50/50 and a wavelength filter before receiving the sequences $S_{A}$($S_{B}$). The PNS can be used to protect against delayed photon attacks by checking for the presence of multi-photon signals, and the quantum wavelength filter removes illegal photons under invisible photon attacks. This setup effectively resists the Trojan horse attack.

### 3.2. The Measure-Resend Attack

Eve intercepts and measures the sequence $S_{B}$ sent by Charlie_n_ to Bob, then prepares and sends fake qubits to Bob based on her measurement results. This allows Eve to steal Bob’s bit string $M_{B}$ without being detected. However, in Step 3 of the protocol, Bob randomly selects a sufficient number of particles from $S_{B}$ to serve as decoy particles for the second security check and shuffles their order. Consequently, Eve cannot determine the exact positions of the decoy particles or the required measurement bases. When considering the polarization DOF, Eve randomly selects either $Z_{P}$ or $X_{P}$ for measurement. The probability that Eve remains undetected is calculated as: $\frac{1}{2}\times 1+\frac{1}{2}\times \frac{1}{2}=\frac{3}{4}$. In a system involving both polarization and spatial-mode DOF, the final probability of detection is: $1-{\left(\frac{3}{4}\right)}^{2d}$, where d represents the number of decoy particles. When d is sufficiently large, the error rate will exceed a predefined threshold, ensuring detection. Based on the above discussion, the protocol effectively defends against the measure-resend attack.

### 3.3. The Entangle-Measure Attack

The entangle-measure attack refers to the scenario where Eve uses a unitary operation $\hat{E}$ to entangle an auxiliary particle ${\left|\varepsilon \right\rangle }_{e}$, which she has prepared, with the intercepted sequence of $S_{B}$ particles. Subsequently, when Bob completes his encoding and transmits the particles to Alice, Eve intercepts them again and applies another unitary operation $\hat{F}$. Finally, by measuring her auxiliary particle ${\left|\varepsilon \right\rangle }_{e}$, Eve attempts to infer useful information.(5)$$\hat{E}=\left(\begin{array}{cc} \alpha_{1} & \beta_{2}\\ \beta_{1} & \alpha_{2} \end{array}\right)⨂\left(\begin{array}{cc} {\alpha_{1}}^{\prime } & {\beta_{2}}^{\prime }\\ {\beta_{1}}^{\prime } & {\alpha_{2}}^{\prime } \end{array}\right)$$

Assuming the initial state of the system is ${|\left.\Phi \right\rangle }_{AB}^{PS}={|\left.\varphi^{+}\right\rangle }^{P}⊗{|\left.\varphi^{+}\right\rangle }^{S}$, the composite state of the system after Eve performs the entanglement operation becomes:$$\hat{E}{|\left.\varphi^{+}\right\rangle }^{P}⊗{|\left.\varphi^{+}\right\rangle }^{S}{\left|\varepsilon \right\rangle }_{e}=\frac{1}{\sqrt{2}}{{[\left|H\right\rangle }_{A}\left(\alpha_{1}{\left|H\right\rangle }_{B}{\left|\varepsilon_{00}\right\rangle }_{e}+\beta_{1}{\left|V\right\rangle }_{B}{\left|\varepsilon_{01}\right\rangle }_{e}\right)+{\left|V\right\rangle }_{A}(\beta_{2}{\left|H\right\rangle }_{B}{\left|\varepsilon_{10}\right\rangle }_{e}+\alpha_{2}{\left|V\right\rangle }_{B}{\left|\varepsilon_{11}\right\rangle }_{e})]}^{P}$$$$⊗\frac{1}{\sqrt{2}}{{[\left|a_{1}\right\rangle }_{A}\left({\alpha_{1}}^{\prime }{\left|b_{1}\right\rangle }_{B}{\left|\varepsilon_{00}\right\rangle }_{e}+{\beta_{1}}^{\prime }{\left|b_{2}\right\rangle }_{B}{\left|\varepsilon_{01}\right\rangle }_{e}\right)+{\left|a_{2}\right\rangle }_{A}({\beta_{2}}^{\prime }{\left|b_{1}\right\rangle }_{B}{\left|\varepsilon_{10}\right\rangle }_{e}+{\alpha_{2}}^{\prime }{\left|b_{2}\right\rangle }_{B}{\left|\varepsilon_{11}\right\rangle }_{e})]}^{S}$$$$=\frac{1}{\sqrt{2}}{{[\left|H\right\rangle }_{B}\left(\alpha_{1}{\left|H\right\rangle }_{A}{\left|\varepsilon_{00}\right\rangle }_{e}+\beta_{2}{\left|V\right\rangle }_{A}{\left|\varepsilon_{10}\right\rangle }_{e}\right)+{\left|V\right\rangle }_{B}(\beta_{1}{\left|H\right\rangle }_{A}{\left|\varepsilon_{01}\right\rangle }_{e}+\alpha_{2}{\left|V\right\rangle }_{A}{\left|\varepsilon_{11}\right\rangle }_{e})]}^{P}$$$$⊗\frac{1}{\sqrt{2}}{[\left|b_{1}\right\rangle }_{B}\left({\alpha_{1}}^{\prime }{\left|a_{1}\right\rangle }_{A}{\left|\varepsilon_{00}\right\rangle }_{e}+{\beta_{2}}^{\prime }{\left|a_{2}\right\rangle }_{A}{\left|\varepsilon_{10}\right\rangle }_{e}\right)+{\left|b_{2}\right\rangle }_{B}({\beta_{1}}^{\prime }{\left|a_{1}\right\rangle }_{A}{\left|\varepsilon_{01}\right\rangle }_{e}$$(6)$${+{\alpha_{2}}^{\prime }{\left|a_{2}\right\rangle }_{A}{\left|\varepsilon_{11}\right\rangle }_{e})]}^{S}$$

${\left[\left|\varepsilon_{00}\right\rangle ,\left|\varepsilon_{01}\right\rangle ,\left|\varepsilon_{10}\right\rangle ,\left|\varepsilon_{11}\right\rangle \right]}^{P/S}$ belongs to the Hilbert space of Eve′s probe states. Clearly, these states must satisfy the condition:$\;{\hat{E}}^{†}\hat{E}=\hat{E}{\hat{E}}^{†}=I$.$${\left[\langle \varepsilon_{00}\left|\varepsilon_{00}\right\rangle +\langle \varepsilon_{01}\left|\varepsilon_{01}\right\rangle \right]}^{P/S}=1$$$${\left[\langle \varepsilon_{10}\left|\varepsilon_{10}\right\rangle +\langle \varepsilon_{11}\left|\varepsilon_{11}\right\rangle \right]}^{P/S}=1$$(7)$${\left[\langle \varepsilon_{00}\left|\varepsilon_{10}\right\rangle +\langle \varepsilon_{01}\left|\varepsilon_{11}\right\rangle \right]}^{P/S}=0$$

Since $\hat{E}$ represents Eve’s unitary operation, it must satisfy the following equation:$${\left|\alpha_{i}\right|}^{2}+{\left|\beta_{i}\right|}^{2}=1,\;{\left|\alpha_{1}\right|}^{2}={\left|\alpha_{2}\right|}^{2}\mathrm{and}\;{\left|\beta_{1}\right|}^{2}={\left|\beta_{2}\right|}^{2}$$(8)$${\left|{\alpha_{i}}^{\prime }\right|}^{2}+{\left|{\beta_{i}}^{\prime }\right|}^{2}=1,\;{\left|{\alpha_{1}}^{\prime }\right|}^{2}={\left|{\alpha_{2}}^{\prime }\right|}^{2}\mathrm{and}\;{\left|{\beta_{1}}^{\prime }\right|}^{2}={\left|{\beta_{2}}^{\prime }\right|}^{2}$$① In Step 3, after Bob measures the target particles to be encoded using the measurement basis $Z_{M}=Z_{P}⊗Z_{S},$ the composite system collapses into a state ${|\left.\Phi^{\prime }\right\rangle }_{ABe}^{PS}$. This state is one of the following possibilities:$${{[\left|H\right\rangle }_{B}\left(\alpha_{1}{\left|H\right\rangle }_{A}{\left|\varepsilon_{00}\right\rangle }_{e}+\beta_{2}{\left|V\right\rangle }_{A}{\left|\varepsilon_{10}\right\rangle }_{e}\right)]}^{P}⊗{{[\left|b_{1}\right\rangle }_{B}\left({\alpha_{1}}^{\prime }{\left|a_{1}\right\rangle }_{A}{\left|\varepsilon_{00}\right\rangle }_{e}+{\beta_{2}}^{\prime }{\left|a_{2}\right\rangle }_{A}{\left|\varepsilon_{10}\right\rangle }_{e}\right)]}^{S}$$$${{[\left|H\right\rangle }_{B}\left(\alpha_{1}{\left|H\right\rangle }_{A}{\left|\varepsilon_{00}\right\rangle }_{e}+\beta_{2}{\left|V\right\rangle }_{A}{\left|\varepsilon_{10}\right\rangle }_{e}\right)]}^{P}⊗{\left[{\left|b_{2}\right\rangle }_{B}({\beta_{1}}^{\prime }{\left|a_{1}\right\rangle }_{A}{\left|\varepsilon_{01}\right\rangle }_{e}+{\alpha_{2}}^{\prime }{\left|a_{2}\right\rangle }_{A}{\left|\varepsilon_{11}\right\rangle }_{e})\right]}^{S}$$$${\left[{\left|V\right\rangle }_{B}(\beta_{1}{\left|H\right\rangle }_{A}{\left|\varepsilon_{01}\right\rangle }_{e}+\alpha_{2}{\left|V\right\rangle }_{A}{\left|\varepsilon_{11}\right\rangle }_{e})\right]}^{P}⊗{{[\left|b_{1}\right\rangle }_{B}\left({\alpha_{1}}^{\prime }{\left|a_{1}\right\rangle }_{A}{\left|\varepsilon_{00}\right\rangle }_{e}+{\beta_{2}}^{\prime }{\left|a_{2}\right\rangle }_{A}{\left|\varepsilon_{10}\right\rangle }_{e}\right)]}^{S}$$(9)$${\left[{\left|V\right\rangle }_{B}\left(\beta_{1}{\left|H\right\rangle }_{A}{\left|\varepsilon_{01}\right\rangle }_{e}+\alpha_{2}{\left|V\right\rangle }_{A}{\left|\varepsilon_{11}\right\rangle }_{e}\right)\right]}^{P}⊗{\left[{\left|b_{2}\right\rangle }_{B}({\beta_{1}}^{\prime }{\left|a_{1}\right\rangle }_{A}{\left|\varepsilon_{01}\right\rangle }_{e}+{\alpha_{2}}^{\prime }{\left|a_{2}\right\rangle }_{A}{\left|\varepsilon_{11}\right\rangle }_{e})\right]}^{S}$$When Bob completes his encoding and sends the particles to Alice, Eve intercepts them again and applies another unitary operation $\hat{F}$. The state is then updated to:(10)$$\hat{F}{|\left.\Phi^{\prime }\right\rangle }_{ABe}^{PS}{\left|\delta \right\rangle }_{e}={|\left.\Phi^{\prime \prime }\right\rangle }_{ABe}^{PS}$$Specifically:$${{\hat{F}[\left|H\right\rangle }_{B}\left(\alpha_{1}{\left|H\right\rangle }_{A}{\left|\varepsilon_{00}\right\rangle }_{e}+\beta_{2}{\left|V\right\rangle }_{A}{\left|\varepsilon_{10}\right\rangle }_{e}\right)]}^{P}⊗{{[\left|b_{1}\right\rangle }_{B}\left({\alpha_{1}}^{\prime }{\left|a_{1}\right\rangle }_{A}{\left|\varepsilon_{00}\right\rangle }_{e}+{\beta_{2}}^{\prime }{\left|a_{2}\right\rangle }_{A}{\left|\varepsilon_{10}\right\rangle }_{e}\right)]}^{S}$$$$=[{\left.\alpha_{1}{\left|H\right\rangle }_{A}{{\left|H\right\rangle }_{B}\left|\delta_{00}\right\rangle }_{e}+\beta_{2}{\left|V\right\rangle }_{A}{{\left|H\right\rangle }_{B}\left|\delta_{10}\right\rangle }_{e}\right.]}^{P}$$(11)$$⊗{\left[\left.{\alpha_{1}}^{\prime }{\left|a_{1}\right\rangle }_{A}{{\left|b_{1}\right\rangle }_{B}\left|\delta_{00}\right\rangle }_{e}+{\beta_{2}}^{\prime }{\left|a_{2}\right\rangle }_{A}{{\left|b_{1}\right\rangle }_{B}\left|\delta_{10}\right\rangle }_{e}\right.\right]}^{S}$$If Eve aims to execute a perfect entangle-measure attack without being detected by the legitimate parties, the state in Equation (11) will collapse into the following form:(12)$$[{\left.{\left|H\right\rangle }_{A}{{\left|H\right\rangle }_{B}\left|\delta_{00}\right\rangle }_{e}\right.]}^{P}⊗{\left[\left.{\left|a_{1}\right\rangle }_{A}{{\left|b_{1}\right\rangle }_{B}\left|\delta_{00}\right\rangle }_{e}\right.\right]}^{S}$$
with the conditions:(13)$$\alpha_{1}={\alpha_{1}}^{\prime }=1,\beta_{2}={\beta_{2}}^{\prime }=0$$

Similarly, for the remaining three cases in Equation (9), analogous results can be obtained, such as:$$\alpha_{1}={\alpha_{2}}^{\prime }=1,\beta_{2}={\beta_{1}}^{\prime }=0$$$$\alpha_{2}={\alpha_{1}}^{\prime }=1,\beta_{1}={\beta_{2}}^{\prime }=0$$(14)$$\alpha_{2}={\alpha_{2}}^{\prime }=1,\beta_{1}={\beta_{1}}^{\prime }=0$$② If Eve intercepts the Reflect particles and applies two unitary operations, $\hat{E}$ and $\hat{F}$, the state of the system will undergo transformations as follows:$$\hat{E}\hat{F}{|\left.\varphi^{+}\right\rangle }^{P}⊗{|\left.\varphi^{+}\right\rangle }^{S}{\left|\varepsilon \right\rangle }_{e}=\frac{1}{\sqrt{2}}{{[\left|H\right\rangle }_{A}\left(\alpha_{1}{\left|H\right\rangle }_{B}{\left|\delta_{00}\right\rangle }_{e}+\beta_{1}{\left|V\right\rangle }_{B}{\left|\delta_{01}\right\rangle }_{e}\right)+{\left|V\right\rangle }_{A}(\beta_{2}{\left|H\right\rangle }_{B}{\left|\delta_{10}\right\rangle }_{e}+\alpha_{2}{\left|V\right\rangle }_{B}{\left|\delta_{11}\right\rangle }_{e})]}^{P}$$$$⊗\frac{1}{\sqrt{2}}{[\left|a_{1}\right\rangle }_{A}\left({\alpha_{1}}^{\prime }{\left|b_{1}\right\rangle }_{B}{\left|\delta_{00}\right\rangle }_{e}+{\beta_{1}}^{\prime }{\left|b_{2}\right\rangle }_{B}{\left|\delta_{01}\right\rangle }_{e}\right)+{\left|a_{2}\right\rangle }_{A}({\beta_{2}}^{\prime }{\left|b_{1}\right\rangle }_{B}{\left|\delta_{10}\right\rangle }_{e}$$(15)$${{{+\alpha }_{2}}^{\prime }{\left|b_{2}\right\rangle }_{B}{\left|\delta_{11}\right\rangle }_{e})]}^{S}$$It also satisfies Equations (8), (13), and (14):$$\hat{E}\hat{F}{|\left.\varphi^{+}\right\rangle }^{P}⊗{|\left.\varphi^{+}\right\rangle }^{S}{\left|\varepsilon \right\rangle }_{e}=\frac{1}{\sqrt{2}}{{(\left|H\right\rangle }_{A}\left.{\left|H\right\rangle }_{B}{\left|\delta_{00}\right\rangle }_{e}\right.+{\left|V\right\rangle }_{A}{\left|V\right\rangle }_{B}{\left|\delta_{11}\right\rangle }_{e})}^{P}$$(16)$$⊗\frac{1}{\sqrt{2}}{{(\left|a_{1}\right\rangle }_{A}\left.{\left|b_{1}\right\rangle }_{B}{\left|\delta_{00}\right\rangle }_{e}\right.+{\left|a_{2}\right\rangle }_{A}{\left|b_{2}\right\rangle }_{B}{\left|\delta_{11}\right\rangle }_{e})}^{S}$$From the transformation of Equation (2), we know that:$${\left|HH\right\rangle }^{P}=\frac{1}{\sqrt{2}}({\left|\varphi^{+}\right\rangle }^{P}+{\left|\varphi^{-}\right\rangle }^{P})$$$${\left|VV\right\rangle }^{P}=\frac{1}{\sqrt{2}}({\left|\varphi^{+}\right\rangle }^{P}-{\left|\varphi^{-}\right\rangle }^{P})$$$${\;\left|a_{1}b_{1}\right\rangle }^{P}=\frac{1}{\sqrt{2}}({\left|\varphi^{+}\right\rangle }^{S}+{\left|\varphi^{-}\right\rangle }^{S})$$(17)$${\left|a_{2}b_{2}\right\rangle }^{P}=\frac{1}{\sqrt{2}}({\left|\varphi^{+}\right\rangle }^{S}-{\left|\varphi^{-}\right\rangle }^{S})$$Substituting Equation (17) into Equation (16), we obtain:$$\hat{E}\hat{F}{|\left.\varphi^{+}\right\rangle }^{P}⊗{|\left.\varphi^{+}\right\rangle }^{S}{\left|\varepsilon \right\rangle }_{e}=\frac{1}{2}{\left[{\left|\varphi^{+}\right\rangle }^{P}(\left.{\left|\delta_{00}\right\rangle }_{e}\right.+{\left|\delta_{11}\right\rangle }_{e}\right)}^{P}+{\left|\varphi^{-}\right\rangle }^{P}{\left(\left.{\left|\delta_{00}\right\rangle }_{e}\right.-{\left|\delta_{11}\right\rangle }_{e}\right)}^{P}]$$(18)$$⊗\frac{1}{2}{\left[{\left|\varphi^{+}\right\rangle }^{S}(\left.{\left|\delta_{00}\right\rangle }_{e}\right.+{\left|\delta_{11}\right\rangle }_{e}\right)}^{S}+{\left|\varphi^{-}\right\rangle }^{S}{\left(\left.{\left|\delta_{00}\right\rangle }_{e}\right.-{\left|\delta_{11}\right\rangle }_{e}\right)}^{S}$$Similarly, for Eve to avoid detection during this entangle-measure attack, Equation (18) must satisfy the following conditions:(19)$$\left.{{\left|\delta_{00}\right\rangle }_{e}}^{P}\right.={{\left|\delta_{11}\right\rangle }_{e}}^{P},\left.{{\left|\delta_{00}\right\rangle }_{e}}^{S}\right.={{\left|\delta_{11}\right\rangle }_{e}}^{S}$$

In summary, it is not difficult to conclude that, regardless of whether the attack occurs in the polarization DOF or the spatial-mode DOF, Eve cannot distinguish between the states ${\left|\varepsilon_{ij}\right\rangle }^{P/S}$. This means that Eve cannot obtain any useful information. Therefore, the protocol demonstrates robustness against the entangle-measure attack. Figure 3 shows a schematic diagram of the entangle-measure attack.

### 3.4. Attacks by the Dishonest Controller

Compared to external attacks initiated by an eavesdropper like Eve, internal attacks by a dishonest controller are more severe. In the MCSQD system, the controller acts as a core component responsible for coordinating the communication process among participants. If the controller behaves dishonestly, it poses a significant threat to the overall security of the communication system. The following analysis explores specific methods of attack by a dishonest controller.

#### 3.4.1. Entanglement Substitution

A dishonest controller Charlie_1_ does not prepare the agreed-upon hyperentangled Bell state ${|\left.\Phi \right\rangle }_{AB}^{PS}={|\left.\varphi^{+}\right\rangle }^{P}⊗{|\left.\varphi^{+}\right\rangle }^{S}$ but instead uses a different hyperentangled state ${\left|\Phi \right\rangle }_{ABC}^{PS}$.(20)$${\left|\Phi \right\rangle }_{ABC}^{PS}=\frac{1}{\sqrt{2}}{\left(\left|HHH\right\rangle +\left|VVV\right\rangle \right)}_{ABC}^{P}⊗\frac{1}{\sqrt{2}}{\left(\left|a_{1}b_{1}c_{1}\right\rangle +\left|a_{2}b_{2}c_{2}\right\rangle \right)}_{ABC}^{S}$$
Subsequently, Charlie_n_ sends the sequence $S_{A}$ to Alice and $S_{B}$ to Bob. Charlie_n_ then attempts to steal Alice and Bob’s secret messages by measuring the sequence $S_{C}$ using the measurement basis. However, in Step 3, Bob prepares new qubits based on the string to be transmitted and uses a permutation operator to shuffle the order of the particles. This shuffling prevents Charlie_n_ from knowing the quantum states of the encoded Code particles. Consequently, Charlie_n_ cannot deduce Bob’s secret information $M_{B}$.

#### 3.4.2. Operation Tampering

A malicious controller Charlie_i_ could deliberately broadcast erroneous unitary operations $R_{1}$ to disrupt the communication process. If Alice does not recover the correct initial state due to this misinformation, it will result in discrepancies during subsequent measurements, preventing her from correctly comparing the outcomes with the initial system state. If Charlie_i_ publishes an erroneous unitary operation $R_{1}$, Alice will be unable to accurately restore the initial hyperentangled state prepared by Charlie_1_. During the verification phase, when Alice and Charlie_n_ compare their measurement results, the QBER will be higher than the pre-defined threshold due to the mismatch caused by the incorrect operation. This high QBER indicates potential tampering or errors in the protocol, leading to the termination of the protocol. Consequently, Charlie_i_ will not be able to obtain any useful secret messages from Alice or Bob.

#### 3.4.3. Deliberate Disclosure

When compromised controllers (e.g., Charlie_i_) collude with external attacker Eve, they may leak operational parameters R to facilitate secret theft. The attack proceeds as follows: Eve intercepts particle sequence $S_{A}$ sent from Charlie_n_ to Alice, forging a counterfeit sequence $S_{A}\prime$ to transmit to Alice. If all controllers are untrustworthy and disclose their operations $\left\{R_{i}\right\}$ to Eve, she may deduce the initial system state ${|\left.\Phi \right\rangle }_{AB}^{PS}$ through reverse engineering to extract secrets. In Step 2 of the protocol, Alice selects sufficient particles from the remaining N−n received sequences for security verification. Since Eve learns the system state after sending $S_{A}\prime$, she cannot predict Alice’s chosen measurement basis. This inevitably triggers a QBER exceeding the predefined security threshold, forcing communication termination. As a result, the collusion between the controller and Eve will inevitably be detected.

### 3.5. Attack by the Dishonest Party

Typically, an attack by a dishonest participant, also known as a collusion attack, involves two parties attempting to bypass the control imposed by the controller, aiming to communicate directly without the controller’s permission. However, in the proposed MCSQD protocol, before the hyperentangled state sequences are distributed, the A particles in sequence $S_{A}$ undergo unitary operations R={I,X,H} by multi-party controllers to encode control information. Therefore, without the controller’s permission, the encoded particles lose their original correlations, making it impossible for the participants to deduce the correct secret information.

## 4. Efficiency Analysis

Quantum communication systems rely on rare and expensive quantum resources. Improving efficiency means transmitting more information while maintaining the same level of resource consumption, thereby optimizing resource utilization and minimizing costs. The information-theoretic efficiency defined in reference [34] is given by:(21)$$\eta =\frac{c}{q+b}$$
where c represents the number of bits used to transmit secret messages, q denotes the total number of qubits employed in the protocol, and b indicates the number of classical bits consumed. Notably, decoy qubits employed for eavesdropping detection are excluded from this calculation. As Feng et al. [35] demonstrated that when the number of decoy particles reaches 20, the probability of detecting an eavesdropper approaches 100%. In the proposed protocol’s large-scale data transmission, decoy particle quantities exhibit negligible impact on total efficiency and are therefore omitted from the efficiency analysis. The total number of qubits consumed in the protocol is q=4n, which includes: n pairs of hyperentangled photon prepared by Charlie_1_, and 2n new separable single-photon qubits prepared by Bob in Step 3. During the transmission process, Bob needs to announce the correct order of the permutation operators, which consumes 2n classical bits. Therefore, b=2n. The protocol successfully transmits c=4n secret message bits. Substituting these values into Equation (21), the efficiency of the protocol is calculated as: $\eta =\frac{4n}{4n+2n}=66.7\%$.

A comparison of the proposed MCSQD protocol with related SQD protocols is shown in Table 2. It is evident that the proposed protocol achieves a relatively high communication efficiency. Moreover, it is suitable for a wide range of complex scenarios and offers significantly richer functionality compared to the SQD protocols listed in the table.

## 5. Conclusions

This study presents a novel multi-party controlled semi-quantum dialogue protocol based on hyperentangled Bell states. Within the protocol, Alice and Bob first apply classical data compression techniques to reduce redundancy. Subsequently, they employ quantum state encoding methods to map the compressed information onto hyperentangled Bell states within the polarization-spatial mode DOF. The multi-party controllers implement hierarchical authorization, permitting communication between participants only when all controllers concur on authorization. The introduction of Hoffman compression encoding, especially for the device-constrained classical party, can greatly save the physical storage space of the participants. Security analysis demonstrates that the protocol is robust against both common external and internal attacks posed by dishonest participants or controllers. Compared to existing SQD protocols, the proposed MCSQD protocol exhibits significant advantages in terms of resource conservation and transmission efficiency. Although the generation of hyperentangled states presents some challenges, the protocol remains feasible with current technology, thereby offering promising prospects for practical applications.

## Figures and Tables

**Figure 1 entropy-27-00666-f001:**
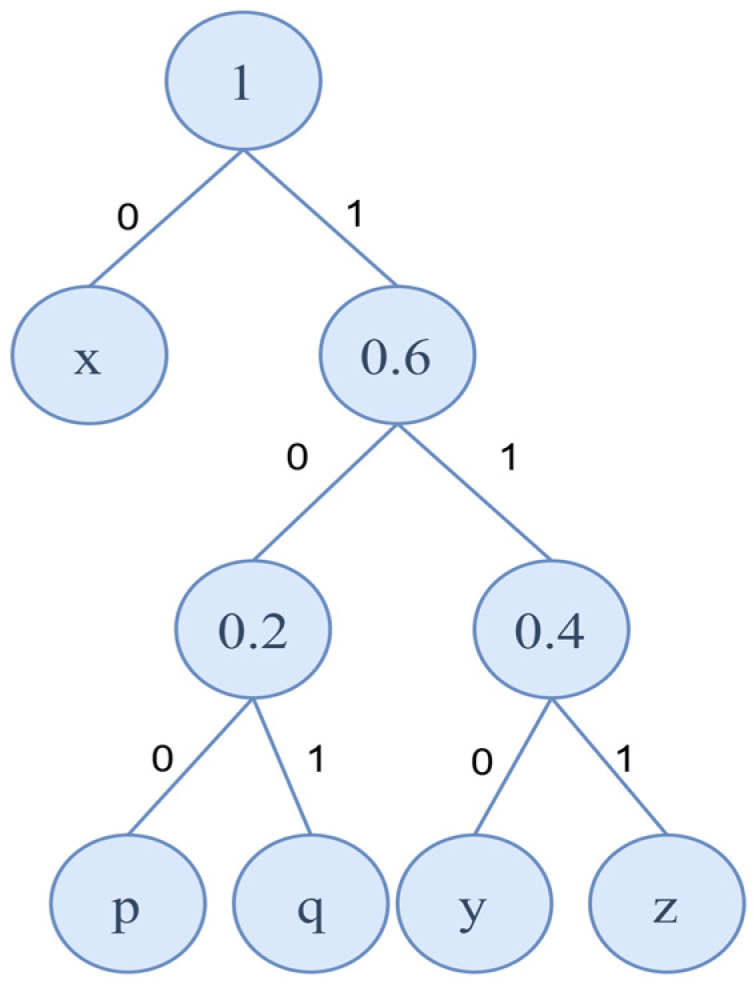
Example of a Huffman Tree Diagram.

**Figure 2 entropy-27-00666-f002:**
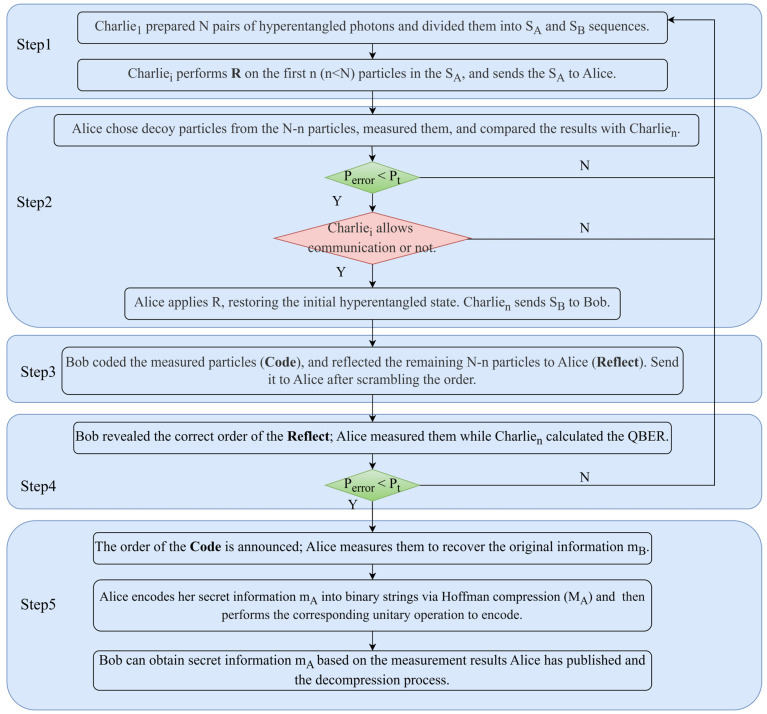
Schematic diagram of the MCSQD protocol.

**Figure 3 entropy-27-00666-f003:**
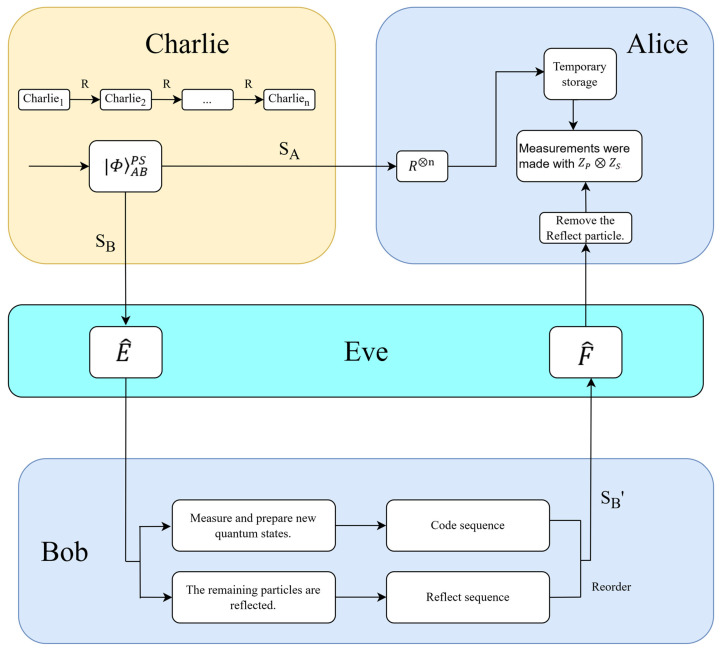
Schematic diagram of the entangle-measure attack.

**Table 1 entropy-27-00666-t001:** Binary codes corresponding to each character.

Character Values	Binary Encoding
x	0
y	110
z	111
p	100
q	101

**Table 2 entropy-27-00666-t002:** Efficiency comparison.

Protocols	The Quantum State Used	c	q	b	η
Shukla [21]	Bell States	2n	3n	3n	50%
Zhou [36]	GHZ State	3n	5n	0	60%
Pan [37]	Bell States	2n	4n	2n	33.3%
Shi [23]	Hyperentangled Bell States	2n	3n	n	50%
Shi [38]	Bell States	2n	4n	n	50%
MCSQD	Hyperentangled Bell States	4n	4n	2n	66.7%

## Data Availability

Data are contained within the article.

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
