# Peer review of "Multi-Party Controlled Semi-Quantum Dialogue Protocol Based on Hyperentangled Bell States"

_entropy, 2025, doi:10.3390/e27070666_

Round 1
Reviewer 1 Report
Comments and Suggestions for Authors
Regarding the manuscript titled Multi-Party Controlled Semi-Quantum Dialogue Protocol Based on Hyperentangled Bell States by Meng-Na Zhao et al., which addresses the critical dual challenges of excessive classical resource consumption and security vulnerabilities in semi-quantum dialogue systems. The authors innovatively propose a multi-party controlled protocol leveraging polarization-spatial hyperentangled Bell states. This protocol achieves quantum channel controllability through hierarchically loading control information by controllers, integrated with an adaptive Huffman compression algorithm and an immediate measurement-transmission mechanism. These features significantly enhance resource efficiency and security robustness.
To further strengthen the manuscript, the following revisions are recommended according to the paper’s structure:
1.Abstract: Clarify the multi-controller architecture (e.g., number of controllers, hierarchical relationships).
2.Introduction: Supplement recent literature (last 2 years) in relevant fields.
3.Protocol Section: Elaborate on the Huffman compression encoding (include key concepts such as its binary tree structure).
4.Efficiency Analysis:Incorporate cross-comparative case studies to quantify and accentuate the protocol's competitive advantages.
In summary, this work demonstrates substantial innovation. The above revisions constitute non-substantive enhancements. We recommend the authors implement these refinements to further amplify the academic value of this research, providing a significant technical reference for the quantum communications community.
Reviewer 2 Report
Comments and Suggestions for Authors
This paper proposes an innovative solution to the two major problems of classical resource consumption and security vulnerabilities in the field of semi-quantum communication. The authors combine Hoffman compression coding with polarization- spatial hyperentangled Bell states to reduce resource consumption and improve transmission efficiency. And leverage multiple controllers to increase the controllability and security of the protocol. However, the protocol is lacking in conceptual rigor and academic normativeness, and it needs to focus on the following issues before it can be published:
- This paper emphasizes "multi-party control" and "Huffman compression coding" as innovative points, but does not fully demonstrate the differences between them and existing schemes. Although this paper extends to multi-party control (Charlie1 to Charlien) for the introduction of third-party control in the semi-quantum protocol, it does not analyze the improvement of protocol security by increasing this operation. It is important to add that there is a need for multiple controllers, not a simple stack.
- In the abstract, there is a misunderstanding of the phrase “An "Immediate measurement and transmission" mechanism is introduced.” Because judging from the current literature, "Immediate measurement and transmission" is not the first time that this paper has proposed it.
- Bob, as a classical party, does not have full quantum capabilities and can only perform partial operations, which the author does not declare in the original text. It is suggested that the text include what Bob can only do to better understand the content for readers who are not familiar with the field.
- In Section 4, there is an error in the efficiency calculation of this protocol, and the classical Bob needs to consume 2n qubits to prepare a separable single-photon state.
- In Section 5, " In this protocol, Alice and Bob employ Huffman compression coding to encode information onto hyperentangled Bell states in polarization-spatial mode DOF." There is a conceptual error in this statement. Hoffman compression coding is a lossless data compression algorithm that reduces the redundancy of information, which is mainly suitable for the compression of classical information, rather than for the coding of quantum states.
- Assuming that some of the multiple controllers are untrustworthy (e.g., Charlie1 colludes with Eve), the protocol relies on the controllers to operate and publish R in turn, but does not analyze the impact of tampering with or leaking R by the untrusted controllers on the system. It is necessary to supplement the QBER calculation and coping strategies in this scenario.
